# Radiotherapy for HER 2 Positive Brain Metastases: Urgent Need for a Paradigm Shift

**DOI:** 10.3390/cancers14061514

**Published:** 2022-03-15

**Authors:** Edy Ippolito, Sonia Silipigni, Paolo Matteucci, Carlo Greco, Sofia Carrafiello, Vincenzo Palumbo, Claudia Tacconi, Claudia Talocco, Michele Fiore, Rolando Maria D’Angelillo, Sara Ramella

**Affiliations:** 1Radiation Oncology, Campus Bio-Medico University, Via Alvaro del Portillo 21, 00128 Rome, Italy; e.ippolito@policlinicocampus.it (E.I.); s.silipigni@policlinicocampus.it (S.S.); c.greco@policlinicocampus.it (C.G.); s.carrafiello@unicampus.it (S.C.); v.palumbo@unicampus.it (V.P.); c.tacconi@unicampus.it (C.T.); c.talocco@unicampus.it (C.T.); m.fiore@policlinicocampus.it (M.F.); s.ramella@policlinicocampus.it (S.R.); 2Radiation Oncology, Tor Vergata University, 00133 Rome, Italy; d.angelillo@med.uniroma2.it

**Keywords:** brain metastases, stereotactic radiotherapy, HER2+, multimodal treatment, radiosensitization, side effects

## Abstract

**Simple Summary:**

Brain metastases (BMs) are common among patients with advanced HER2 breast cancer. The recent introduction of systemic therapy with central nervous system activity as well as the wider use of brain stereotactic radiotherapy (SRT) are contributing to improving the outcomes for these patients. In this review, we discuss a modified approach to the treatment of HER2-positive BMs from a radiation oncologist point of view, taking into consideration new advances in multimodal therapy and combinations of the most commonly used systemic treatments and brain radiation therapy (RT).

**Abstract:**

Brain metastases (BMs) are common among patients affected by HER2+ metastatic breast cancer (>30%). The management of BMs is usually multimodal, including surgery, radiotherapy, systemic therapy and palliative care. Standard brain radiotherapy (RT) includes the use of stereotactic radiotherapy (SRT) for limited disease and whole brain radiotherapy (WBRT) for extensive disease. The latter is an effective palliative treatment but has a reduced effect on brain local control and BM overall survival, as it is also associated with severe neurocognitive sequelae. Recent advances both in radiation therapy and systemic treatment may change the paradigm in this subset of patients who can experience long survival notwithstanding BMs. In fact, in recent studies, SRT for multiple BM sites (>4) has shown similar efficacy when compared to irradiation of a limited number of lesions (one to three) without increasing toxicity. These findings, in addition to the introduction of new drugs with recognized intracranial activity, may further limit the use of WBRT in favor of SRT, which should be employed for treatment of both multiple-site BMs and for oligo-progressive brain disease. This review summarizes the supporting literature and highlights the need for optimizing combinations of the available treatments in this setting, with a particular focus on radiation therapy.

## 1. Introduction

Breast cancer is the second most common malignancy associated with increased risk of developing brain metastases (BMs). The incidence of BMs in breast cancer (BC) patients has increased over the past decade. Breast cancer brain metastases (BCBMs) occur in approximately 10–16% of breast cancer patients, especially among patients with HER-2-positive breast cancer [1,2,3].

Before the introduction of trastuzumab, a monoclonal antibody that binds to human epidermal growth factor receptor 2 (HER2), the overall survival (OS) in HER2-positive patients was shorter due to the progression of systemic disease. After the introduction of trastuzumab, OS has been significantly improved due to extracranial control of the disease. More recently, due to the introduction of novel HER2-targeted therapies, the survival of HER2-positive BC patients has further increased, ranging from 12 to 24 months [4,5].

Local treatment of BCBM is multimodal and can include surgery, whole brain radiation (WBRT) and stereotactic radiosurgery (SRS) or fractionated stereotactic radiotherapy (fSRT). Surgery is performed in selected cases and should be preferred for larger and symptomatic lesions. Surgery alone is not sufficient for local control, and postoperative radiotherapy (RT) is recommended [6,7,8].

Historically, the standard treatment for patients with multiple BMs has been WBRT, while SRS was applied only to patients with one or a limited number of BMs (two to four). More recently, Yamamoto et al. reported that there was no difference in median OS between patients with 2–4 BMs and 5–10 BMs treated with SRS. In this study, 10% of patients had breast cancer [9,10]. Moreover, the incidence rate of one or more treatment-related adverse events did not differ between patients with 2–4 or 5–10 BMs.

Furthermore, novel anti-HER2 drugs, such as neratinib, tucatinib and trastuzumab deruxtecan, showed relevant central nervous system (CNS) activity in these patients, which adds to the known action of lapatinib-containing regimens [11].

For all the above reasons, and also due to the longer survival experienced by patients with HER2 BCBMs, the management of these patients represents a unique scenario requiring a multidisciplinary and tailored approach. In this review, we will summarize the supporting literature and discuss the option of a modified approach to the treatment of HER2-positive BMs from a radiation oncologist perspective, taking into consideration new advances in multimodal therapy and combinations of the most commonly used systemic treatments and RT.

## 2. Brain Radiotherapy: Lessons from the Past and Novel Treatments

Over the years, brain radiotherapy has continued to evolve thanks to advances in radiation techniques including the use of intensity-modulated radiotherapy (IMRT), volumetric-modulated arc therapy (VMAT) and SRT (Figure 1).

### 2.1. WBRT

WBRT has been used for many years as the mainstay radiation treatment for BMs. The typical dose and fractionation schedule for WBRT is 30 Gray (Gy) in 10 fractions or, alternatively, 20 Gy in 5 fractions. WBRT after either surgical resection or radiosurgery does not improve OS [12,13], while as a palliative treatment it is very effective: 7 out of 10 patients receive benefits in terms of symptom improvement. However, the local control rate using WBRT is limited. Chougule et al. [14] reported a local control of 87%, 91% and 62% for patients treated with SRS (Gamma Knife—GK), SRS + WBRT, and WBRT alone, respectively, suggesting comparatively lower local tumor control for patients undergoing WBRT only.

In order to increase the rates of local control, novel techniques have been developed using a conformational fractionated external beam boost, thereby reaching local control rates above 75%. Dose escalation can be achieved through the use of WBRT + SIB, as well as through the application of WBRT followed by SRS (WBRT + SRS) [15].

The control of dose escalation can be managed with IMRT or VMAT through the use of the simultaneous integrated boost (SIB) technique [16]. Through the use of WBRT + SIB, a boost on BMs can be achieved in a single session with an optimized dose distribution, and the single simulation protocol allows for reduced waiting and dose delivery time. From a radiobiological point of view, SIB techniques reduce the possibility of reoxygenation and re-assortment between fractions. The chance of tumor cell repopulation or a sublet repair process decreases with a reduced time break between WBRT and SRS [17].

Progression outside or in the boost area is strictly related to the applied technique. The incidence of progression outside the dose boost area was found to be significantly lower in patients treated with WBRT + SIB compared with those treated with WBRT + SRS (39.4% vs. 75%). This may be due to the faster reduction in dose outside the BMs in the WBRT + SRS group than in the WBRT + SIB group. The incidence of progression in the dose-boost area was significantly higher in the WBRT + SIB group (60.6%) than in the WBRT + SRS group (25%). This could be due to the higher biologically effective dose (BED) to the P-GTV in the WBRT + SRS group than in the group treated with WBRT + SIB [18].

WBRT is associated with several side effects, such as temporary increase of brain edema and/or hair loss, but mainly neurocognitive function (NCF) impairment [19], that can negatively impact quality of life (QoL) [20]. The most common neurocognitive disfunction is short-term memory loss [21]. NCF decrease can occur 3–6 months after WBRT and can be irreversible and progressive [22].

Since the pathological process behind NCF decline has been mainly attributed to the exposure of neural stem cells of the hippocampus dentate gyrus to radiation, the hippocampal-avoidance technique (HA-WBRT) has been developed in order to reduce NFC decline [23]. This can be feasible, as there is low risk of metastases and recurrence in the hippocampal region [24].

A phase II multi-institutional trial demonstrated that HA-WBRT helps preserve neurocognitive function and quality of life compared with historical controls (*p* < 0.001) [25]. The phase III NRG Oncology CC001 study demonstrated a lower risk of neurocognitive failure after HA-WBRT plus memantine compared with WBRT plus memantine (adjusted hazard ratio, 0.74; 95% CI, 0.58–0.95; *p* = 0.02). Furthermore, HA-WBRT showed no differences in intracranial progression-free survival (PFS), OS and toxicity [26]. Moreover, HA-WBRT without memantine showed better memory preservation at 6-month follow-up compared with WBRT alone [27].

Other strategies to improve NCF include the use of neuroprotective drugs, such as memantine, an antagonist of the N-methyl-D-aspartate (NMDA) receptor used in the setting of vascular dementia. RTOG 0614 aimed to investigate the protective effects of memantine on cognitive function in patients receiving WBRT. In this randomized study, patients enrolled in the memantine arm experienced significantly later onsets of cognitive decline (hazard ratio 0.78, 95% confidence interval 0.62–0.99, *p* = 0.01) [28].

With available technology, it is also possible to specifically and simultaneously deliver fractionated radiation doses to tumors and at-risk brain areas while sparing hippocampal structures [29].

In a study by Westover, a single-arm phase II trial, WBRT (20 Gy in 10 fractions over 2–2.5 weeks) were used together with a simultaneous integrated boost (40 Gy in 10 fractions to identified metastatic lesions) and a hippocampal sparing approach. Patients were then evaluated for NCF decline after treatment using the HVLT-R DR (Hopkins Verbal Learning Test-Revised Delayed Recall), and the results showed significantly better outcomes in patients treated with HSIB-WBRT (hippocampal sparing integrated boost WBRT) compared with historical outcomes and with historical outcomes in patients treated with non-Hippocampal-sparing WBRT and comparable outcomes with those of patients treated with SRS alone; the intracranial control rates were also similar to patients treated with WBRT + SRS [30].

### 2.2. SRS and Fractionated Stereotactic Radiotherapy (fSRT)

SRS is a technique based on the use of intersected beams that can be delivered in both single RT or fSRT. Compared with WBRT, it allows for a more precise target irradiation to a specific area in order to obtain, through the delivery of a highly conformal and high dose of radiation, an ablative effect on target tissue. Its precise targeting allows for a reduction of radiation exposure to the area surrounding the lesion, thereby reducing the damage to adjacent healthy tissue [31].

SRS treatment can, however, be associated with late toxicity characterized mainly by the occurrence of radionecrosis (RN) [32]. RN results from direct brain damage or from necrosis due to an inflammatory cascade mediated by vascular endothelial injury. It can be asymptomatic or it can cause several neurological symptoms, such as seizures, cognitive deficits, headaches and vomiting. The etiology of RN is multifactorial and can depend on RT dose, fractionation, the volume of BMs, the site of irradiation and the volume of healthy brain irradiated. In the literature, the reported incidence rate ranges from 3% to 24% [32].

According to the RTOG 90–05 protocol, the rates of RN observed in patients with recurrent BMs undergoing a new course of brain RT with single-fraction radiosurgery were 8% at 12 months and 11% at 24 months [33].

Increased rates of RN can be observed if SRS is delivered concurrently with systemic therapies. However, the effective incidence cannot be properly quantified as its definition as study endpoint has been inconsistent. Moreover, no imaging modality is the gold standard test to discriminate between RN and tumor progression, even if the ongoing research is testing the role of PET-based or MRI-based imaging [32].

SRS has recently also been employed to treat multiple BMs. VMAT is a new technique used for linac-based SRS or fSRT, which allows, through the use of a single plan with one isocenter, the treatment of multiple targets simultaneously [34,35]. The use of non-coplanar arcs and the simultaneous variation of MLC leaf positions, dose rate, and gantry rotation speed during the treatment dose delivery allows VMAT to be effective in distributing a highly conformational dose [34]. Treating multiple targets at the same time leads to shortened treatment time, possibly reducing the adverse dosimetric effects linked to intrafraction motion [36].

The difference between WBRT and SRS in patients with five or more BMs still remains unclear. A phase III randomized trial compared SRS with WBRT in 5–15 BMs [9]. The trial has demonstrated that there is no difference in the median OS of 10.4 months in the SRS group and 8.4 months in the WBRT group (*p* = 0.45). Thus, the authors concluded that avoiding WBRT may also be possible for patients with more than five metastases [9]. Furthermore, neurocognitive function did not differ between the two groups [10]. Two currently ongoing studies are investigating quality of life differences between HA-WBRT and SRS in patients with 5–20 BMs (NCT03075072 study) and the control rate for 1–10 BMs using fSRT (NCT04061408 study).

## 3. Combination of Brain RT with Systemic Therapy in the Treatment of BCBMs

### 3.1. Brain RT + Trastuzumab

Trastuzumab is the humanized form of the murine antibody directed to the external domain of HER2. Trastuzumab inhibits the growth of a variety of cancer cells overexpressing HER2 and accrues significant clinical benefit in the metastatic and adjuvant settings.

Preclinical studies demonstrated the critical role played by trastuzumab as a radiosensitizer. In fact, the exposure of breast cancer cells with high levels of HER2 to trastuzumab markedly increased the apoptosis rate caused by RT. Moreover, in cells with experimentally elevated levels of HER2, downregulation of HER2 by trastuzumab restored cellular radiosensitivity [37].

Despite the limited blood–brain permeability, several studies demonstrated that trastuzumab alone for the treatment of HER2-positive breast cancer results in a prolonged time to BMs and longer survival time after the diagnosis of BMs [38]. Moreover, even though trastuzumab is a large molecule, studies on 64Cu-DOTA-trastuzumab or 89Zr-trastuzumab PET imaging showed some BBB penetration and metastatic brain lesions in HER2-positive breast cancer patients [39].

Chargary et al. investigated the outcome of 31 patients treated with WBRT and trastuzumab and reported a radiological response in 74.2% and clinical response in 87.1%. No Grade 2 or higher acute toxicity was observed [40]. Miller et al., in a large retrospective series, including 187 HER2+ patients treated with SRS to 374 BCBMs, reported a longer median survival with the concurrent use of HER2 antibodies (17.9 months vs. 15.1 months; *p* = 0.04). Nearly all HER2 antibodies were treated with trastuzumab alone (94%). The reported toxicity was acceptable, with a rate of RN of 3.8% [41]. See Table 1 for details.

### 3.2. Brain RT + Trastuzumab–Pertuzumab

Pertuzumab is a humanized monoclonal antibody which is directed against the extracellular domain of the HER2 receptor and inhibits the interaction of HER2 with HER3. Similar to trastuzumab, pertuzumab stimulates both antibody-dependent and cell-mediated cytotoxicity. In preclinical HER2-positive models, the dual blockade showed increased antitumor effect compared with either agent alone. Dual blockade is also likely to be a powerful radiosensitizer in anti-HER2 therapy. Even if pertuzumab is a large molecule, studies using positron emission tomography–computed tomography (PET-CT) with 89Zr-pertuzumab showed some blood–brain barrier permeability [42]. In the phase II single arm PATRICIA trial, the combination of trastuzumab and pertuzumab achieved an ORR of 11% with a 6-month clinical benefit of 51% in 40 patients progressing after radiotherapy [43].

Not many data are available on the association between dual blockade with pertuzumab and trastuzumab and brain RT. In the phase III CLEOPATRA trial, 23 patients received RT to the brain and only 5 SRS in the pertuzumab–trastuzumab arm [44]. Bergen et al. reported the outcome of 252 HER2-positive BC patients. Among these patients, only 26 were treated with dual blockade and 14 were eligible for response. Of these, 12 patients underwent local therapy (either surgery or WBRT or SRS). The ORR and iCBR were 92.9% and 100.0%, respectively. The only patient who did not receive local therapy (surgery, WBRT or SRT) reported a partial response (PR) [45].

Recently, our institution reported a case series of 10 patients treated with fSRT to 32 BMs. The ORR was 68.7%, while only 1 RN was reported with a median follow-up of 18.3 months. Overall BM median survival was 33.9 months (95% CI 24.1–43.6). The mean duration of dual blockade in these patients was 27.9 months (range: 10.1–53.7 months) [46]. See Table 1 for details.

**Table 1 cancers-14-01514-t001:** Combination of systemic therapy and brain RT: trastuzumab, trastuzumab–pertuzumab.

Authors	Study	Year	Population	Treatment	Outcome	Adverse Effects
Chargari et al. [42]	Retrospective	2011	- 31 patients	- 25 patients: WBRT 30 Gy/10 fr - 6 patients: WBRT other fractionation - 17 patients: trastuzumab 2 mg/kg - Weekly - 14 patients: 6 mg/kg repeated every 21 days	**OR**: 74.2% **Clinical Response**: 87.1%	**RN** **/** **OTHER** **Related to WBRT**: Grade 1 -Headache 8 (26%) -Nausea/vomiting 4 (13%) **Related to trastuzumab** Allergic-like reaction 2 (6.5%) Decrease in LVEF 3 (9.5%)
Miller et al. [43]	Retrospective	2017	- 187 patients (374 lesions)	- 36% upfront SRS - 78% upfront WBRT - 15% underwent upfront surgery - 83% received hormone therapy - 80% received HER2 antibodies - 38% received HER2/EGFR TKIs - 63% received cytotoxic chemotherapy	**OS** Concurrent HER2 antibodies 17.9 months vs. local therapy alone 15.1 months (*p* = 0.04)	**RN** - SRS alone = 5.6% - SRS + trastuzumab = 3.8% - SRS + lapatinib = 1.3%
Ippolito E et al. [47]	Retrospective	2021	- 10 patients (32 lesions)	- fSRT Median dose 27 Gy (range 12–27 Gy)	**OR** 68.7% **iCB** 100% **DIF** 4/10 patients (40.0%) **Median OBMS** 33.9 months (95%CI 24.1–43.6) **Mean duration of PT treatment** 27.9 months (range: 10.1–53.7 months)	**RN** - - 1/32 = 3.1%
				- All concurrent pertuzumab and trastuzumab		

ABBREVIATIONS: WBRT = whole brain radiotherapy; SRS = stereotactic radiosurgery; fSRT = fractionated stereotactic radiotherapy; OR = overall response rate; iCB = intracranial clinical benefit; DIF = distant intracranial failure DIF; OBMS = overall brain metastases survival.

### 3.3. Brain RT + Lapatinib

Lapatinib ditosylate is a dual receptor TKI targeting simultaneously two members of the HER family of receptors, HER1 (EGFR1/ErbB1) and HER2/c-neu (ErbB2), that acts by reversibly binding to the cytoplasmic ATP-binding site of the tyrosine kinase domain, producing the inhibition of various downstream signaling cascades involved in anti-proliferative and pro-apoptotic effects [48].

A small molecule can cross the blood–brain barrier and can therefore reach BMs. In a study using radiolabelled lapatinib, the positron emission PET scan showed selective radioactivity uptake in HER2+ brain metastasis compared with normal brain tissue [47]. Morikawa provided evidence for drug uptake of lapatinib in a surgical series of resected, mostly non-irradiated BCBMs [49]. In a clinical setting, in patients not pretreated with RT, the upfront use of lapatinb resulted in an objective response rate of 66% and progression-free survival of 5.5 months [50].

Furthermore, lapatinib has demonstrated in an in vitro experiment that it can increase radiosensibility in HER2+ breast cancer cells. In particular, lapatinib improved the radiosensitivity of breast SKBR3 and BT474 cells, thereby hindering the repair of DNA damage, as suggested by the prolongation of radiation-induced γH2AX foci and the downregulation of phosphorylated DNA-dependent protein kinase and the catalytic subunit (pDNAPKcs) shown using a clonogenic assay [51]. Lapatinib can also potentiate the radiation-induced irreversible arrest of cell proliferation by increasing expression of caspase 3 and the rate of apoptosis and senescence [52].

In a phase 1 trial aimed at evaluating the maximum tolerated dose (MTD) and the feasibility of lapatinib being given concurrently with WBRT, Lin et al. reported, out of 28 eligible patients, an encouraging CNS objective response rate (ORR) of 79%, with 46% of patients remaining progression-free (CNS or non-CNS) at 6 months [53].

In a large retrospective cohort of 132 HER2+ BCBMs, Kim et al. showed how lapatinib given concurrently with SRS resulted in an improved complete response (35% vs. 11%) and the best objective response (median 100% vs. 70% reduction) compared with SRS alone [54].

In a subsequent cohort of 126 patients with a total of 157 HER2+ BCBMs treated at the same institution with SRS and lapatinib, Parsai et al. reported that the efficacy of combining lapatinib and SRS was greater for smaller lesions, with a greater reduction of 12 months local failure compared with SRS alone (5.7% vs. 15.1%). Moreover, it was unexpectedly noted that the rate of radiation necrosis was also reduced when SRS was given concurrently with lapatinib (1.3% vs. 6.3%) [55].

In a recent meta-analysis, including six studies with 843 HER2 + BCBMs, SRS in association with concurrent lapatinib was also investigated. In two studies, increased survival was reported with the association of SRS and any use of lapatinib (Shireen et al.: 27.3 vs. 19.5 months, *p* = 0.03; Kim et al.: 33.3 vs. 23.6 months, *p* = 0.009). Local control was also significantly increased with SRS plus lapatinib based on the meta-analysis of three studies (HR 0.47 [0.33, 0.66], *p* = 0.0001), in particular when lapatinib was given concurrently. In addition, two studies reported a lower radiation necrosis rate with concurrent lapatinib and SRS [56]. See Table 2 for details.

### 3.4. Brain RT + TDM1

Trastuzumab emtansine (T-DM1) is an immunoconjugated drug consisting of the monoclonal antibody trastuzumab bound to emtansine, a potent maitansin-derived microtubule inhibitor.

Even if it is well demonstrated that HER2 overexpression induces radioresistance, in preclinical studies, T-DM1 failed to induce radiosensitivity in human breast cancer cells expressing HER2. In fact, T-DM1 induced cellular death due to the intracellular action of emtansine. However, the radiation dose needed to achieve 10% cell survival did not decrease when radiotherapy was combined with T-DM1 [57]. In the exploratory analysis of the KAMILLA trial evaluating the safety and efficacy of T-DM1 in 398 patients with stable or occult brain metastases, the intracranial response rate evaluable in 126 patients was 21.4%, with an intracranial clinical benefit of 42.9%. Among 67 patients with measurable BMs not treated with brain RT, a volume reduction >30% was achieved in 49% of patients [59].

Some studies have tried to evaluate the efficacy and safety of a combined treatment with T-DM1 and SRT (see Table 3) [59]. In the majority of these studies, T-DM1 administration in association with brain RT appears to be linked with an increased risk of RN.

One of the first published studies described the outcome of a total of 13 breast cancer patients receiving SRS, of which 7 were also treated with T-DM1: 57% of patients treated with SRS + T-DM1 developed RN [65].

In another small series of 12 patients (PS = 0–1) with HER2+ BCBMs treated from 2014 to 2015 with T-DM1 (every 3 weeks) and concurrent or sequential radiosurgery with or without WBRT, a radiological response was obtained in three out of four cases (75%), with PD in 25% of patients (5.5 months for local progression and 6.5 months for brain progression). In the arm where the sequential SRS + T-DM1 approach was combined, the RN rate was 33.3%, with an edema rate of 28.6%, and 50% in the concurrent group, with an edema rate of 25% [66].

Stumpf et al. analyzed the safety profile of T-DM1 and SRS in a total of 45 patients with BCBMs treated between 2014 and 2017. Of these, 30/45 patietns were HER2-positive. Out of 45 patients, 23 received T-DM1, of which 16 received T-DM1 with SRS/FSRT (7 T-DM1 + sequential SRS–6 T-DM1 + concurrent SRS), while 22/45 received T-DM1 alone. Approximately 22% of patients were found to develop clinically significant radionecrosis, especially in the combined approach, with concomitant SRS of 23 patients who received T-DM1 + RT (vs. 1 out of 22 patients who did not receive T-DM1). The authors additionally provided pre-clinical evidence of the increased rate of RN, explaining how unintended T-DM1 targeting of reactive astrocytes in healthy nearby brain tissue is a mechanism underlying T-DM1/SRS-induced toxicity [60].

More recently, Mills et al. reported on the outcome of 16 patients with HER2 + BCBMs who received SRS or fSRT with concomitant or sequential T-DM1 from December 2013 to December 2019. Overall, 40 BCBMs were treated, of which 24 were treated with SRS (median dose 21 Gy) and 16 with fSRT (median dose 25 Gy delivered in 3 to 5 fractions). Moreover, 19 metastases were treated with RT + concurrent T-DM1, 11 with RT delivered before T-DM1, 10 with RT delivered after T-DM1. After SRS or FSRT, local control (LC) and distant intracranial control (DIC) at 20 months was 75% and 50%, respectively. Moreover, OS at 20 months was 67% after SRS or FSRT and 79% following diagnosis of BCBMs. In this series, only 1 case of radionecrosis was reported (179 days after combined therapy with T-DM1 and SRS on 5 lesions, with the first dose 8 days before SRT). Furthermore, in 8/18 treatment sessions, mild radiation-related side effects were reported (such as headaches and fatigue) [61].

One study, published in abstract form only, showed that T-DM1 exposure post-SRS was independently associated with a higher risk of RN in HER2+ BMBC patients. There were 67 patients with HER2 + BMBC (223 lesions) treated with SRS identified; among these, 21 (31.3%) were treated with post-SRS T-DM1, with a median follow-up of 15.6 months. Considering independent risk factors of RN, either T-DM1 treatment post-SRS (HR 2.5, 95% CI 1.2–5.3, *p* = 0.02) or an RT dose with a BED >50.4 Gy (HR 2.4, 95% CI 1.1–5.1, *p* = 0.02) was given; patients treated with T-DM1 and SRS had a 25.2% (95% CI 12.8–37.6%) risk of RN at both 1- and 2-years post-T-DM1, with 80% of all RN cases occurring within 12 months of T-DM1 treatment [62]. See Table 3 for details.

### 3.5. Limitations

Evaluating the safety and efficacy of combined treatment is difficult and limited by the retrospective nature of the available studies. Moreover, as more new drugs with intracranial activity are going to be used in the near future, combination with RT will be more complex. In fact, in order to extract information that could add to current clinical practice in this area, it would be useful to include data on brain radiotherapy treatments (i.e., type, dose, interval time from delivery of RT to the beginning of systemic therapy). However, most clinical trials available do not provide such data.

On the other hand, only a few prospective studies of combined treatments are ongoing (see Table 4).

Finally, regarding the occurrence of RN, the main limitation is related to the absence of a consistent defined endpoint across combination studies. In fact, different grades of RN are often reported together, not even distinguishing symptomatic from asymptomatic events. This can be crucial, as symptomatic RN is less common but clinically meaningful.

## 4. Future Developments: How the Therapeutic Algorithm Should Be Reshaped

With this knowledge of all the advances in HER2+ BMs treatment, the optimal treatment algorithm for these patients cannot be the same as that used to treat BMs from other malignancies. However, a strategy based on patient performance status and symptoms, the intracranial volume of disease, the burden of the extracranial disease and the planned systemic therapy could define the best approach for both first BM occurrence or subsequent relapse (see Figure 2 and Figure 3).

### 4.1. When Should WBRT Be Delivered?

WBRT for newly diagnosed BMs should be delivered in very selected cases as a palliative treatment or to increase intracranial control in patients with symptomatic disease or disease rapidly progressing after brain SRT. See Table 5 for details.

When WBRT is the indicated treatment, clinical and technical approaches to reduce NCF impairment should always be used for patients with a life expectancy >6 months.

### 4.2. When Should SRS/fSRT Be Delivered?

SRS/fSRT should always be the first RT option for patients with HER2+ BCBMs. Recent guidelines recommend the use of this treatment option for patients with 1–4 metastases [63]. However, in this setting, patients with multiple BMs could experience long survival [64,67,68]. For this reason, once a patient has been accurately selected (good PS, extracranial disease controlled, first occurrence), SRT/fSRT should only be used after a multidisciplinary discussion.

SRS/fSRT should also be used for late relapse (>6 months) after a first RT treatment course (either WBRT or SRS/fSRT). It can also be used as a boost dose to larger BMs after WBRT, taking into account the increased risk of RN.

### 4.3. When Can RT Deferral Be an Option?

RT deferral can be a valid option for patients with diffuse asymptomatic BMs who are eligible for a systemic therapy with intracranial activity, providing close MRI follow-up evaluations are undertaken to properly treat progression. Recent guidelines recommend this approach for patients to be treated with tucatinib combination therapy [68], even though lapatinib combinations may play a role in this setting.

### 4.4. How to Manage the Association of Systemic Therapy with Brain RT?

Figure 4 summarizes the rate of RN reported in the studies investigating the association between SRT and the most commonly used systemic therapies in HER2 patients.

Trastuzumab, as well as trastuzumab together with pertuzumab, could be administered concurrently with RT, even though there are not many data available on the association with brain RT. With regard to T-DM1, due to the high frequency of brain radionecrosis reported in retrospective studies, concurrent administration of T-DM1 and brain SBRT should be avoided, and, if necessary, strategies to reduce the rate of RN should be employed (for instance, preferring fSRT or hypofractionated SRT, especially for larger lesions, to limit the volume of healthy brain receiving high RT doses). As retrospective studies suggest, the association of a low frequency of brain RN with the concurrent administration of lapatinib and brain RT should be considered as the safest option to treat a large number of metastases and larger lesions. However, given the lack of clinical comparative data, it is difficult to evaluate whether the combination of lapatinib and brain RT would be more effective than novel TKIs, such as neratinib or tucatinib. Moreover, to the best of our knowledge, no data are available on the combination with brain RT for novel drugs, such as new generation TKIs, as well as trastuzumab deruxetan in combination with brain RT. Therefore, patients undergoing these therapies should not be treated concurrently with brain RT and side effects should be carefully monitored.

## Figures and Tables

**Figure 1 cancers-14-01514-f001:**
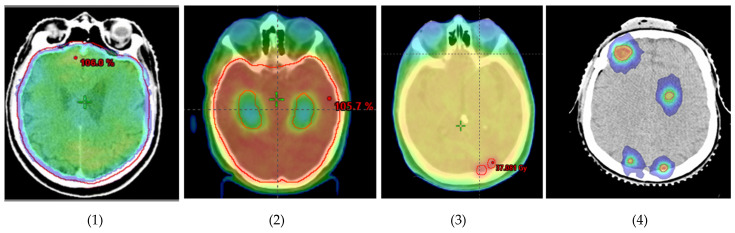
Evolution of brain radiotherapy (from left to right): (1) WBRT; (2) HA-WBRT; (3) WBRT + SIB; (4) multiple-site SRT.

**Figure 2 cancers-14-01514-f002:**
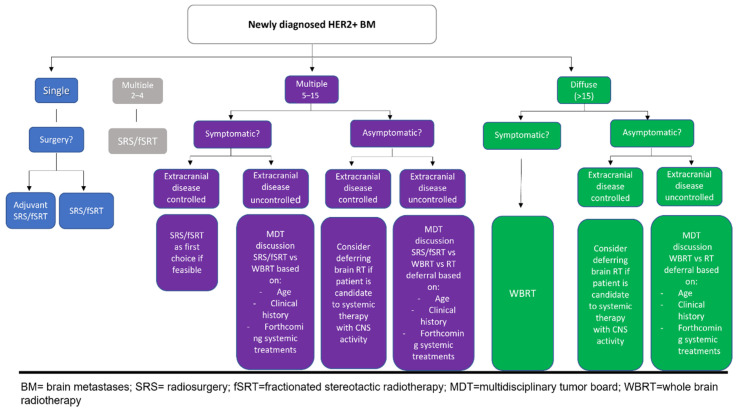
Proposed algorithm for newly diagnosed HER2 BMs.

**Figure 3 cancers-14-01514-f003:**
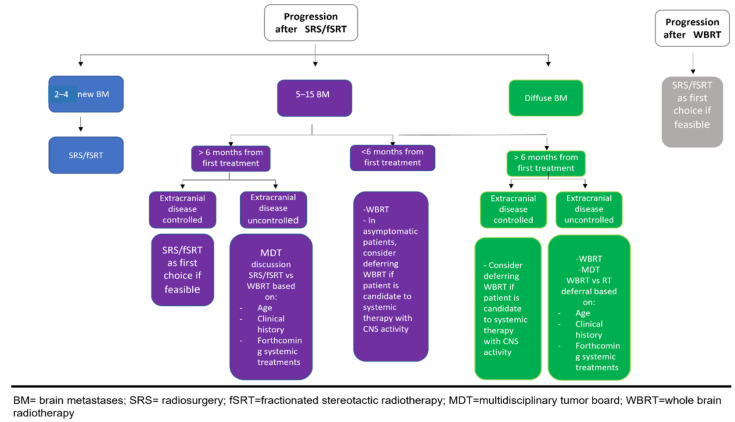
Proposed algorithm for recurrent HER2 BMs after brain RT.

**Figure 4 cancers-14-01514-f004:**
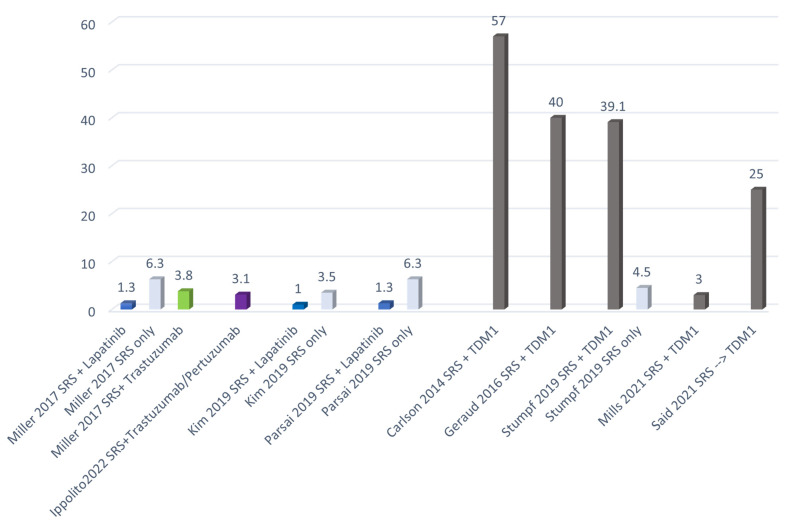
RN incidence reported in studies investigating the association between SRT and systemic therapies.

**Table 2 cancers-14-01514-t002:** Combination of systemic therapy and brain RT: lapatinib.

Authors	Study	Year	Population	Treatment	Outcome	Adverse Effects
Lin et al. [55]	Phase I	2013	- 28 patients	- WBRT (37.5 Gy/15 fr) - Lapatinib Day 1, a loading dose of 750 mg for two doses. Day 2, three DLs once daily: 1000 (DL1), 1250 (DL2), and 1500 (DL3) mg	**ORR** 79% **6 mPFS** 46% at 6 months	**RN** /
Parsai et al. [57]	Retrospective	2019	- 126 patients (479 lesions)	- 24 patients concurrent lapatinib + SRS - SRS +/− WBRT	**LF** - Concurrent lapatinib + SRS 12-month (5.7% vs. 15.1%, *p* < 0.01), 24-month (6.5% vs. 18.0%, *p* < 0.01)	**RN** - 1.3% cohort treated with lapatinib - SRS alone 6.3% (*p*< 0.01)
Kim et al. [56]	Retrospective	2019	- 84 patients (487 lesions)	SRS (18–24 Gy)+/−WBRT - 132 lesions (27%) SRS + lapatinib - 355 lesions (73%) only SRS	**CR** - Lapatinib + SRS 35% - Only SRS 11% (*p* = 0.008) **OR** - Lapatinib + SRS -100% - Only SRS - 70% (*p* < 0.001)	**RN** - SRS + lapatinib 1.0% - Only SRS 3.5% (*p* = 0.27)
M Khan et al. [58]	Review–meta-analysis	2020	- 6 studies - 843 HER2+ patients - SRS +/− WBRT - WBRT	279 patients: lapatinib +/− anti-HER2 therapy - 610 patients: anti-HER2 therapy (mainly trastuzumab) - 227 patients: no anti-HER2 therapy	**OS** - Lapatinib-based therapy was associated with increase in **OS** (HR 0.63 *p* < 0.00001) **LC** significantly increased with concurrent lapatinib + SRS (HR 0.47 (0.33, 0.66), *p* = 0.0001)	**RN** Lower rate for concurrent lapatinib + SRS: - Miller: 1.3% vs. 6.3% (*p* 0.001) - Kim: 1% vs. 3.5% (*p* = 0.134) - Parsai: 1.3% vs. 6.3% (*p* = 0.001)

ABBREVIATIONS: SRS = stereotactic radiosurgery; BCBMs = breast cancer brain metastases; ORR = objective response rate; CR = complete response; OR = objective response; WBRT = whole brain radotherapy; LF = local failure; OS = overall survival; LC = local control; RN = Radionecrosis.

**Table 3 cancers-14-01514-t003:** Combination of systemic therapy and brain RT: TDM1.

Authors	Study	Year	Population	Treatment	Outcome	Adverse Effects
Carlson J et al. [60]	Retrospective	2014	- 13 patients	7/13 patients T-DM1 + SRS	NR	**RN** 57% (4/7) SRS + T-DM1
Geraud A et al. [61]	Retrospective	2017	- 12 patients	- 4/12 SRS + Concurrent T-DM1 - 8/12 SRS + Sequencial T-DM1	SRS + Concurrent T-DM1 - OR 75% - PD 25% SRS + Sequential T-DM1 - OR 75% - PD 25%	**RN** - 33.3% (4/12) in the sequential group - 50% (2/4) in the concurrent group **OTHER** - Edema rate of 28.6% in the sequential group - Edema rate of 25% in the concurrent group
Stumpf PK et al. [62]	Retrospective	2019	-45 patients	TDM1 + RT: -23/45 (51.1%): -16 SRS/FSRT + Concurrent T-DM1: -7 SRS and sequential T-DM1 - Only SRS 22/45 -15 patients also received WBRT	NR	**RN** 10/45 (22.2%) clinically significant: - 9 of 23 (39.1%) patients received T-DM1 (3 sequential SRS, 6 concurrent SRS) - 1 of 22 (4.5%) patients did not receive T-DM1
Mills MN et al. [63]	Retrospective	2021	- 16 patients (40 lesions)	RT: - 24 SRS (median dose 21 Gy) - 16 FSRT (median dose 25 Gy/3–5 fractions) T-DM1: - 19 RT + Concurrent T-DM1 - 11 RT and sequential T-DM1 - 10 T-DM1 and sequential RT	- 20 months LC 75% (SRS or FSRT) - 20 months DIC 50% (SRS or FSRT) - 20 months OS 67% (SRS or FSRT) and 79% (following BCBM diagnosis)	**RN** 1 patient (3%) **OTHER**: 45% G1–G2 Headaches and fatigue
Id Said B et al. [64]	Retrospective	2021	- 67 patients (223 lesions)	21/67 patients (31.3%) SRS and sequential T-DM1	/	**RN** - Post-SRS 1-year 6.7% (95% CI 2.7–10.7%) 2-year 15.2% (95% CI 9.2–21.3%) - SRS and sequential T-DM1 1- and 2-years 25.2% (95% CI 12.8–37.6%)

ABBREVIATIONS: TDM1 = trastuzumab emtansine; BCBMs = breast cancer brain metastases; fSRT = fractionated stereotactic radiotherapy; SRS = Stereotactic radiosurgery; RT = radiotherapy; LC = local control; OS = overall survival; WBRT = whole brain radiotherapy; OR = overall response rate; PD = progression disease; RN = radionecrosis.

**Table 4 cancers-14-01514-t004:** Ongoing clinical trials of combinations of brain RT with systemic therapy in the treatment of BCBMs.

Clinicaltrials Identifier	Title	Phase	N° Patients	Treatment Arms	Endpoint
NCT05042791	Pyrotinib Combined With Brain Radiotherapy in Breast Cancer Patients With BMs	2 Randomized	362	Experimental: fSRT combined with pyrotinib and capecitabine Active Comparator: WBRT combined with pyrotinib and capecitabine	1-year objective response rate of central nervous system
NCT04582968	A Study of Pyrotinib Plus Capecitabine Combined With SRT in HER2+ MBC With BMs	I/II	47	fSRT or WBRT drug: pyrotinib combined with capecitabine pyrotinib, 400 mg once daily; capecitabine 1000 mg/m^2^ per day on day 1 through 14, every 21 days	1. Safety and tolerability of pyrotinib plus capecitabine combined with brain radiotherapy 2. Intracranial local tumor control rate

ABBREVIATIONS: BMs = brain metastases; fSRT = fractionated stereotactic radiotherapy; WBRT = whole brain radiotherapy.

**Table 5 cancers-14-01514-t005:** Patient candidates for WBRT.

WBRT Intent	Patient Category
Palliative	Patients with a life expectancy of less than 6 months
Increase intracranial control	Patients diagnosed with symptomatic diffuse BMs or multiple BMs (5–15) who are not suitable for SRS/fSRT. Multidisciplinary evaluation is recommended in this latter case for an accurate risk/benefit assessment in terms of disease control and neurocognitive side effects, with particular reference to age, patient preferences and therapeutic options to control extracranial disease
Increase intracranial control	Patients who rapidly progress (<6 months) after a first course of brain RT treatment or with associated systemic progression, with an evaluation of the therapeutic options available on the basis of the patient’s clinical history

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
