# Peer review of "Radiotherapy for HER 2 Positive Brain Metastases: Urgent Need for a Paradigm Shift"

_cancers, 2022, doi:10.3390/cancers14061514_

Round 1

Reviewer 1 Report

The review is well organized and written and the provided information updated.

I would suggest to briefly add two concepts.

  1. 2nd page, WBRT, after the first general statement, I would add a new one such as “WBRT after either surgical resection or radiosurgery does not improve overall survival (Aoyama H, Shirato H, Tago M, et al. Stereotactic radiosurgery plus whole-brain radiation therapy vs. stereotactic radiosurgery alone for treatment of brain metastases: A randomized controlled trial. JAMA 2006;295:2483–2491; Kocher M, Soffietti R, Abacioglu U, et al. Adjuvant whole-brain radiotherapy versus observation after radiosurgery or surgical resection of one to three cerebral metastases: results of the EORTC 22952-26001 study. J Clin Oncol. 2011 Jan 10;29(2):134-41. doi: 10.1200/JCO.2010.30.1655. Epub 2010 Nov 1. PMID: 21041710; PMCID: PMC3058272) while as palliative treatment.
  2. 3rd page, after the statement ending with the reference n.18 I would add "that can negatively impact QoL" (Soffietti R, Kocher M, Abacioglu UM, et al. A European Organisation for Research and Treatment of Cancer phase III trial of adjuvant whole-brain radiotherapy versus observation in patients with one to three brain metastases from solid tumors after surgical resection or radiosurgery: quality-of-life results. J Clin Oncol. 2013 Jan 1;31(1):65-72. doi: 10.1200/JCO.2011.41.0639. Epub 2012 Dec 3. PMID: 23213105).

Author Response

We would like to thank the reviewer for the time spent reviewing our manuscript. We modified the manuscript according to the reviewer’s comments. All changes are tracked in the manuscript and additional section have been highlighted. We also provided a point by point response to the reviewers’comments.

COMMENT 1: 2nd page, WBRT, after the first general statement, I would add a new one such as “WBRT after either surgical resection or radiosurgery does not improve overall survival (Aoyama H, Shirato H, Tago M, et al. Stereotactic radiosurgery plus whole-brain radiation therapy vs. stereotactic radiosurgery alone for treatment of brain metastases: A randomized controlled trial. JAMA 2006;295:2483–2491; Kocher M, Soffietti R, Abacioglu U, et al. Adjuvant whole-brain radiotherapy versus observation after radiosurgery or surgical resection of one to three cerebral metastases: results of the EORTC 22952-26001 study. J Clin Oncol. 2011 Jan 10;29(2):134-41. doi: 10.1200/JCO.2010.30.1655. Epub 2010 Nov 1. PMID: 21041710; PMCID: PMC3058272) while as palliative treatment.

RESPONSE 1: as you suggested we added this notion (2nd page, WBRT paragraph) and added the references in the bibliography.

COMMENT 2: 3rd page, after the statement ending with the reference n.18 I would add "that can negatively impact QoL" (Soffietti R, Kocher M, Abacioglu UM, et al. A European Organisation for Research and Treatment of Cancer phase III trial of adjuvant whole-brain radiotherapy versus observation in patients with one to three brain metastases from solid tumors after surgical resection or radiosurgery: quality-of-life results. J Clin Oncol. 2013 Jan 1;31(1):65-72. doi: 10.1200/JCO.2011.41.0639. Epub 2012 Dec 3. PMID: 23213105).

RESPONSE 1: as you suggested we added this sentence (3rd page, WBRT paragraph) and added the reference in the bibliography.

Reviewer 2 Report

This is a very nicely put review on the recent interventions in Brain metastases (BM) interventions including the use of drug intervention coupled with the use of stereotactic radiotherapy (SRT)and whole-brain radiotherapy (WBRT) that can bring a new paradigm in the treatment of BM.

the review is well written and has the optimal components for the concepts of drug and radiation therapy in combination.

I think that the article would benefit if there are tables that indicate the efficacy of using the individual therapy rather than being in combination.

In addition, the current tables can be enriched by adding more details on the treatment paradigms and length of treatment, etc....

also, a limitation section should be added.

the section 4.1. "When should WBRT be delivered?" should be written as a paragraph and the items can be added as a table.

Author Response

We would like to thank the reviewer for the time spent reviewing our manuscript. We modified the manuscript according to the reviewer’s comments. All changes are tracked in the manuscript and additional section have been highlighted. We also provided a point by point response to the reviewers’comments.

COMMENT 1. I think that the article would benefit if there are tables that indicate the efficacy of using the individual therapy rather than being in combination.

RESPONSE 1. We appreciate your comment and we added in each section a sentence about the intracranial action of the drug alone (page 5, 8). However since the main focus of the review is to address the challenge of combination we did not add tables for systemic therapy alone.

COMMENT 2. In addition, the current tables can be enriched by adding more details on the treatment paradigms and length of treatment, etc....

RESPONSE 2. As you suggested, we added a column “treatment” to the tables giving more details about treatment.

COMMENT 3: also, a limitation section should be added.

RESPONSE 3: as you suggested we added a “limitation section” in the section dedicated to combined treatment (page 11).

COMMENT 4.:the section 4.1. "When should WBRT be delivered?" should be written as a paragraph and the items can be added as a table.

RESPONSE 4: we modified the section 4.1 following your suggestions.

Round 2

Reviewer 1 Report

The manuscript is now fine for me

Author Response

we thank the reviewer for the final comment